# Occupational Exposure to Pesticides Affects Pivotal Immunologic Anti-Tumor Responses in Breast Cancer Women from the Intermediate Risk of Recurrence and Death

**DOI:** 10.3390/cancers14215199

**Published:** 2022-10-23

**Authors:** Janaína Carla da Silva, Thalita Basso Scandolara, Rodrigo Kern, Hellen dos Santos Jaques, Jessica Malanowski, Fernanda Mara Alves, Daniel Rech, Guilherme Ferreira Silveira, Carolina Panis

**Affiliations:** 1Laboratory of Tumor Biology, Center of Health Sciences, State University of West Paraná, Francisco Beltrão 85605-010, Paraná, Brazil; 2Center of Health Sciences, Federal University of Rio de Janeiro, Rio de Janeiro 21941-902, Rio de Janeiro, Brazil; 3Francisco Beltrão Cancer Hospital (Ceonc), Francisco Beltrão 85601-080, Paraná, Brazil; 4Carlos Chagas Institute, Fiocruz/PR, Curitiba 81310-020, Paraná, Brazil

**Keywords:** breast cancer, risk stratification, cytokine, oxidative stress, inflammation, pesticide exposure

## Abstract

**Simple Summary:**

This study presents information regarding the immunological changes induced by pesticide exposure in patients diagnosed with breast cancer occupationally exposed to pesticides. Such changes are helpful to understand tumor behavior under pesticide exposure and can be beneficial to re-stratify breast cancer patients occupationally exposed concerning their risk of disease recurrence and death.

**Abstract:**

Breast cancer risk stratification is a strategy based using on clinical parameters to predict patients’ risk of recurrence or death, categorized as low, intermediate, or high risk. Both low and high risk are based on well-defined clinical parameters. However, the intermediate risk depends on more malleable parameters. It means an increased possibility for either suboptimal treatment, leading to disease recurrence, or systemic damage due to drug overload toxicity. Therefore, identifying new factors that help to characterize better the intermediate-risk stratification, such as environmental exposures, is necessary. For this purpose, we evaluated the impact of occupational exposure to pesticides on the systemic profile of cytokines (IL-12, IL-4, IL-17A, and TNF-α) and oxidative stress (hydroperoxides, total antioxidants, and nitric oxide metabolites), as well as TGF-β1, CTLA-4, CD8, and CD4 expression, investigated in tumor cells. Occupational exposure to pesticides decreased the levels of IL-12 and significantly increased the expression of TGF-β1 and CTLA-4 in the immune infiltrate. Nevertheless, we observed a decrease in CTLA-4 in tumor samples and CD8 in infiltrating cells of intermediate overweight or obese patients with at least one metastatic lymph node at the diagnosis. These findings indicate that occupational exposure to pesticides changes the molecular behavior of disease and should be considered for intermediate-risk stratification assessment in breast cancer patients.

## 1. Introduction

Breast cancer is the leading cause of death in women worldwide [1]. It is a heterogeneous disease with distinct clinical outcomes depending on patients’ characteristics and the tumor’s biology.

Due to its complexity, several approaches guide clinicians in decision-making. In this context, the risk stratification of disease recurrence and death is a helpful strategy based on clinical parameters defined by international guidelines, used to guide therapeutic decisions and predict clinical outcomes, such as response to treatment and survival [2,3,4].

Although the stratification of low- and high-risk patients lies on well-defined parameters, those from the intermediate-risk category represent a challenge, as they present more conditions for categorizing patients than the other sets. This group represents the most significant frustration for clinicians since they cannot accurately differentiate patients who have a micro-metastatic disease and need treatment from those who are cured with local treatment [5]. Therefore, it is clear that other parameters than those customarily used to categorize them need to be included. 

Several parameters are potential candidates that could help strengthen the current risk stratification. Pesticide exposure depicts an embracing environmental risk factor that could be enclosed in the risk stratification of breast cancer patients in areas of severe agrochemical exposure, such as Brazil.

Studies demonstrate that pesticides can modify breast cancer biology by genotoxic effects [6], chromosomal damage [7], inflammation triggering [8], and the induction of oxidative stress [9], all closely linked to tumor development, genomic instability, and disease progression [10,11]. 

Furthermore, pesticide exposure has already been shown to deregulate the inflammatory and immunological axis in occupationally exposed patients diagnosed with breast cancer, including systemic depletion of tumor necrosis factor-alpha (TNF-α) and interleukin-1β (IL-1β), both pivotal anti-tumor cytokines [12]. It is well known that inflammation is a hallmark of cancer [13], and its deregulation by pesticides may bring relevant consequences to tumor progression. Cytokines can act in dual ways, inhibiting tumor development or contributing to the chronic inflammation that supports tumor growth, which has been associated with poor prognosis in cancer patients [14]. Although there are associations between the risk of breast cancer and occupational exposure to pesticides [14,15,16] and its induced endocrine disruption and carcinogenesis [14,15,16,17,18], little is known about its impact on the risk stratification of patients and their tumor biology.

Therefore, the present study investigates whether occupational exposure to pesticides in women with breast cancer affects tumor biology in patients classified as intermediate-risk. For this purpose, we investigated the systemic inflammatory profile by analyzing the levels of tumor necrosis factor-alpha (TNF-α), interleukin 4 (IL-4), interleukin 17A (Il-17A), and interleukin 12 (IL-12), as well as oxidative stress markers and tumor expression of cytotoxic T-lymphocyte-associated antigen 4 (CTLA-4), transforming growth factor-beta 1 (TGF-β1), CD8, and CD4 tumor immune infiltrate. In addition, we correlated these findings with clinicopathological features from patients.

## 2. Materials and Methods

### 2.1. Study Design

The primary collection of data of women treated at the Cancer Hospital of Francisco Beltrão, Paraná, Brazil (CEONC), was carried out between May 2015 and August 2021; 664 patients were screened during this period, and 342 were diagnosed with breast cancer. 

The Institutional Ethics Committee approved this proposal under the number CAAE 35524814.4.0000.0107, and only patients who signed an informed consent form were considered for this study.

The risk stratification of disease recurrence and death was performed according to the following information collected from the medical records: estrogen receptor expression (ER), progesterone receptor expression (PR), human epidermal growth factor 2 receptor amplification (HER2), histological tumor grade, presence of lymph node metastases, tumor size, the molecular subtype of breast cancer, and age at diagnosis. 

Patients’ pesticide occupational exposure profile was obtained based on a systematized instrument containing questions about their working routine, such as time of exposure in years, type of exposure, amount, and chemical classes of pesticides [19]. All patients included in the study were continuously exposed to pesticides, before and after diagnosis, since they were rural workers. 

Thus, based on the combined risk stratification and pesticide exposure data, 130 women were eligible for this study and stratified as intermediate-risk patients occupationally exposed to pesticides (IE, *n* = 77) and intermediate-risk patients not exposed to pesticides (INE, *n* = 53). The study design is shown in Figure 1, and all parameters considered for risk stratification are shown in Table 1.

### 2.2. Sample Obtention and Determination of the Systemic Inflammatory Profile

Heparinized blood samples (5 to 10 mL) were centrifuged for 5 min at 4000 rpm, and the plasma obtained was stored at −20 °C until the analysis. 

As previously described, pro-oxidative profile was determined by measuring the plasmatic hydroperoxide levels using high-sensitivity chemiluminescence (CL) [20]. Aliquots of 125 μL of plasma were added to 855 μL of monobasic phosphate buffer (10 mM, pH 7.4). The reaction was initiated by adding 20 μL of tert-butyl solution (3 mM) and monitored in a Glomax 20/20 luminometer (Promega, Madison, WI, USA). The results were obtained by integrating the area under the curve of photon emission and expressed in relative light units (RLU).

The antioxidant profile was obtained by analyzing the total radical antioxidant parameter (TRAP) [21]. It consisted of adding 50 uL of 20 mM 2,2′-azobis (2-amidinopropane, ABAP) solution and 50 uL of a 40 µM luminol solution in 10 uL of plasma samples diluted 1:50 in a saline buffer. The reaction was monitored in a Glomax luminometer, one read per second for 5 to 30 min. The results were calculated concerning a standard curve using a hydrosoluble form of vitamin E standard (Trolox). Results were expressed as μM of Trolox.

The oxidation index was obtained from the ratio between CL and TRAP and expressed in arbitrary units.

To estimate the levels of nitric oxide metabolites (NOx), aliquots of 60 uL of plasma samples were deproteinized with zinc sulfate (ZnSO4, 75 mM) and 70 μL sodium hydroxide (NaOH 100 mM) solution. After centrifugation at 10,000 rpm for 2 min, the supernatant was recovered, and 50 μL of glycine (NaOH 45 g/L pH 9.7) buffer was added. The nitrate-to-nitrite conversion was performed using the cadmium–copper reaction method, and the detection of total nitrite was performed using Griess reagent, as described [22]. The results are expressed in μM of nitrite.

The cytokines IL-12, IL-4, IL-17A, and TNF-α were quantified using the enzyme immunoassay (enzyme-linked immunosorbent assay (ELISA)) method, using a commercial kit (e-Biosciences, San Diego, CA, USA), according to the manufacturer’s guidelines. Results were calculated in pg/mL from standard curve data for each cytokine. The kit’s detection limit is 2 pg/mL.

### 2.3. Tumor Analysis: Immunofluorescence Labeling for Transforming Growth Factor-β1, Cytotoxic T-Lymphocyte-Associated Antigen 4, and CD4/CD8 Lymphocyte Labeling

TGF-β1, CTLA-4, CD4, and CD8 expression levels in tumor samples were analyzed by immunofluorescence under the conditions presented in Table 2. Poly-lysine-coated slides containing 3 μm thick sections were kept for 24 h in an incubator at 65 °C. Subsequently, a chemical deparaffinization was carried out for 5 min consecutively in xylene, ethanol, and water. Antigen retrieval was performed only for CTLA-4 in humid heat with the immersion of the slides in 1 mM sodium citrate buffer at pH 6.0 for 15 min at 100 °C. TGF-β1, CD4, and CD8 did not require antigenic retrieval because they presented excellent immunostaining without this process. Slides were incubated in a 3% hydrogen peroxide solution for 15 min to block endogenous peroxidase. Sections were washed between each step with saline solution (sodium chloride—150 mM NaCl in pH 7.6) and 1% Tween 20. Blocking of non-specific binding sites was also performed with a 5% powdered milk solution for 15 min. Then, the slides were incubated with the specific primary antibody in a closed humid chamber, as specified in Table 2. Afterward, the slides were washed and usually incubated for 1 h with diluted secondary antibody. The sections were treated with 4,6 diamidinophenylindole (DAPI) (Sigma Aldrich, St. Louis, MO, USA), 5 mg/mL solution, for 30 min for nuclear counterstaining and assembled in glycerol.

Images were captured in a Motic BA410E fluorescence microscope coupled to a MOTICAM ProS5 Plus camera and Motic Images Plus 3.0 ML image acquisition and processing software. Slides were read using a DAPI excitation filter at 200 or 400 × magnification. The following adjustments were predefined: auto exposure, gain + 20, and zero offsets. The images obtained were stored in BMP format, resolution 2048 × 1536 pixels. The Reporting Recommendations for Tumor Marker Prognostic Studies (REMARK) were followed.

### 2.4. Statistical Analysis

Data distribution was tested using the Shapiro–Wilk test. Thus, normally distributed variables were analyzed with parametric tests. When the normality assumption was not met, nonparametric tests were applied. To compare molecular data between the intermediate group occupationally exposed and unexposed to pesticides, Student’s *t*-test or the Mann–Whitney test was used to compare the two groups. The results were analyzed using GraphPad Prism version 7.0 (Graphpad Software, San Diego, CA, USA). The primary data comparative analysis of the risk stratification profile of patients according to occupational exposure to pesticides was performed using the chi-square test for independence. Correlation analysis of clinicopathological data according to systemic levels of IL-12 was performed with the R programming language (R Development Core Team). Data are represented as median, minimum, and maximum in the results description and box-plot graphs.

## 3. Results

Descriptive data and a comparative analysis of the clinicopathological parameters of intermediate-risk patients from pesticide-exposed and unexposed groups are shown in Table 3. The predominant molecular subtype was luminal B for both groups (43% in the IE and 57% in the INE group). Other key characteristics for both groups were: tumor size above 2, histological grade I/II, no lymph node metastases, age above 50 years, Ki67 proliferation index, and menopause at diagnosis. No statistical differences were observed between the groups regarding such features. However, the IE group showed a significant predominance of overweight/obese patients (*p* = 0.04) and at least one affected lymph node at the time of diagnosis (*p* = 0.03). 

The oxidative stress profile (Figure 2) was evaluated by measuring the systemic hydroperoxide level (Figure 2A), TRAP (Figure 2B), oxidation index (Figure 2C), and NOx levels (Figure 2D).

No significant differences were found when comparing IE versus INE patients for hydroperoxide (772.13: 268.25–1904.08 RLU and 823.14: 290.91–2019.60 RLU, respectively, *p* = 0.55), TRAP (358.1: 82.84–1074 nM Trolox for IE and 298.8: 26.93–828.8 nM Trolox for INE, *p* = 0.41), oxidation index (2941: 132.4–9655 arbitrary units for IE and 4226: 143.4–14,777 arbitrary units for INE, *p* = 0.19), and NOx (32.33: 7.19–59.15 μM for IE and 32.07: 6.54–58.93 μM for INE), *p* = 0.89).

For the cytokine profile (Figure 3), IL-4 (Figure 3A), IL-17-A (Figure 3B), and TNF-α (Figure 3C) were evaluated and showed no significant differences (IL-4—36.89: 9.73–117.1 pg/mL for INE and 32.84: 6.46–72.22 pg/mL for IE, *p* = 0.39; IL-17A—71.2: 6.32–222.7 pg/mL for INE and 7765: 13.52–204.1 pg/mL for IE, *p* = 0.57; TNF-α—94.87: 19.38–151 pg/mL for INE and 95.36: 12.21–245 pg/mL for IE, *p* = 0.96).

Nevertheless, IL-12 (Figure 3D) showed a significant decrease in the IE patients compared to those from the INE group (61.32: 5.66–190.9 pg/mL for INE and 27.37: 10.88–49.81 pg/mL for IE, *p* = 0.03).

Spearman’s correlation analysis was performed concerning IL-12 levels and all clinicopathological parameters in both groups (Figure 4). The heat map shows that the stronger correlations (red squares) are located in opposite parameters when comparing IE and INE groups (Figure 4A,B). It is possible to note that the strongest correlations regarding IL-12 levels observed in the IE group are focused on tumor characteristics (hormone receptors, Ki67 index, tumor subtypes). At the same time, INE patients concentrate on patients’ characteristics (BMI, menopause, age at diagnosis). A significant positive correlation was found between overweight/obese patients and positive hormonal receptors for IE patients, R= 0.52 and *p* = 0.002.

Since IL-12 was systemically reduced, we also investigated the expression of molecules described as negative regulators of the immune response, TGF-β1 and CTLA-4 (Figure 5). Tumor-infiltrating cells from the IE group showed significantly increased expression of both markers (TGF-β1, *p* = 0.001 and CTLA-4, *p* = 0.002, Table 4). Meanwhile, in tumor cells from INE patients, a higher expression of CTLA-4 was found concerning IE (*p* = 0.001).

We also labeled CD4 and CD8 lymphocytes in IE and INE tumor samples to characterize the infiltrating cells. The results demonstrated that the proportion of CD4 cells is similar in both groups (*p* > 0.05). However, CD8 infiltrate was significantly reduced in the IE group, displaying positivity in only 30% of all analyzed samples (*p* = 0.0059, Table 4).

## 4. Discussion

This study evaluated the meaning of inflammatory parameters in the clinical context of the risk stratification of recurrence and death and pesticide exposure for breast cancer patients. We observed that pesticides could induce significant changes in immune components, reducing anti-tumor mediators in the bloodstream and enhancing the expression of immune evasion proteins in tumor-infiltrating cells. These findings reinforce the impact of pesticide-induced deregulation on the anti-tumoral immune response in intermediate-risk breast cancer patients, the most undefined clinic condition regarding breast cancer risk stratification for recurrence and death. As far as we know, this information is novel and adds to the literature regarding the importance of anti-tumoral immune response in the context of breast cancer prognostic and therapeutic decisions.

Breast cancer is a disease characterized by an extensive systemic inflammatory response that includes variations in cytokines, immune signaling mediators, and oxidative stress metabolite levels [23,24,25,26]. In particular, investigating patients’ immunological profiles has emerged as a point of interest over the years. 

Activation of immunity against tumors triggers the production of a series of circulating mediators [27], which can act dually, inhibiting tumor development or contributing to the chronic inflammation that supports tumor growth, associated with poor prognosis in cancer patients [14]. 

In this context, we analyzed the circulating levels of cytokines linked to specific T-helper profiles and observed a reduction in IL-12 in the intermediate-risk patients occupationally exposed to pesticides. IL-12 is one of the main anti-tumor interleukins, acting through pleiotropic effects on different immune cells that form the tumor microenvironment, establishing links between innate and adaptive responses [28]. This cytokine can suppress tumorigenesis and induce regression of established tumors, promoting Th1-related adaptive immunity and cytotoxic response. The IL-12 depletion observed in the exposed patients may impair the immune response against tumors, leading to a higher risk of poor disease-related outcomes. In addition, such patients exhibited a concomitant increase in TGF-β1 and CTLA-4 expression in tumors’ infiltrate cells. 

It is known that TGF-β1 negatively regulates IL-12 levels [29] and is linked to immune evasion and poor responses to cancer immunotherapy [30]. In the early stages of the disease, TGF-β1 inhibits epithelial cell cycle progression and promotes apoptosis, showing tumor-suppressive effects. However, in more advanced settings, it is related to tumor progression, cell motility, cancer invasiveness, and metastasis [31,32].

Pesticides can increase TGF-β1 expression and activation in breast cancer cells, enhancing migration and invasion [33]. These findings support a putative mechanism triggered by pesticide exposure, which has pro-tumor effects on intermediate-risk patients by enhancing TGF-β1 expression in breast tumors and depleting their systemic IL-12 levels. 

We also observed high expression of the immune checkpoint regulator CTLA-4 in breast tumor infiltrates from the exposed patients. Literature reports some associations between CTLA-4 expression in the tumor microenvironment and worse outcomes [34], reaching more than 50% of cases [35]. Even though we did not find any specific study about the impact of pesticides on CTLA-4 expression and considering the IL-12 and TGF-β1-induced changes, exposure to such substances significantly affects the negative regulation of the immune environment of breast cancer patients. 

Concerning the profile of tumor-infiltrating cells, despite CD4-positive cells being found in a similar proportion in both groups, CD8-positive cells were significantly reduced in the exposed group. Although there are no literature data about the impact of pesticide exposure on the immune profile of breast tumor infiltrate, pesticide-induced lymphocyte toxicity has been demonstrated by several mechanisms, including reduced cell counting [36], decreased Th1 cells [37], and low CD4+ and CD8+ subpopulations in blood [38]. This shows that pesticide exposure affects the leukocyte infiltrate pattern and impairs cytokine production in breast cancer patients categorized as intermediate-risk for disease recurrence and death when they are occupationally exposed. Evaluating these markers may help refine disease stratification in breast cancer patients under chronic pesticide exposure, as in the Brazilian population. 

Although several pertinent pathways for the effects of xenobiotics have been identified, the mechanisms of action for IL-12, TGF-β1, and CTLA-4 in this context remain to be established. Considering the clinical meaning of our findings, we observed that most patients exhibiting deregulation of the anti-tumor immune response are overweight or obese at the time of diagnosis. In addition, patients with overweight/obesity showed a high correlation with hormonal receptors according to IL-12 levels. Obesity is a well-known risk factor for breast tumor development associated with inferior survival, and the described mechanisms include a complex network formed by inflammation, ROS generation, epigenetic changes, and mitochondrial dysfunction [39,40,41]. Furthermore, obesity can increase the risk of disease recurrence and death by 35 to 40% [42]. A case –control study [43] observed a positive association between obesity and premenopausal breast cancer risk. Thus, understanding what lies behind this is a significant opportunity to improve prognosis, therapy decisions, and outcomes.

Most of these patients also presented one or more affected lymph nodes, implying a greater probability of metastases to distant organs, increasing the risk of disease recurrence and decreasing the overall survival [44,45]. Our results demonstrate that pesticide exposure affects critical immunologic anti-tumor responses in breast cancer women stratified as having an intermediate risk of recurrence, suggesting it as a valuable possible additional risk factor to be further investigated in the risk stratification of recurrence and death protocol.

## 5. Conclusions

As summarized in Figure 6, the present study shows that occupational exposure to pesticides has a meaningful impact on breast cancer patients’ systemic and tumor inflammatory profile from the intermediate risk of recurrence and death. Impairment in systemic IL-12 levels associated with increased expression of TGF-β1 and CTLA-4 in tumor and infiltrated immune cells may represent a putative signature of pesticide-induced immune impairment in breast cancer that should be considered when calculating the risk of recurrence and death of these patients.

## Figures and Tables

**Figure 1 cancers-14-05199-f001:**
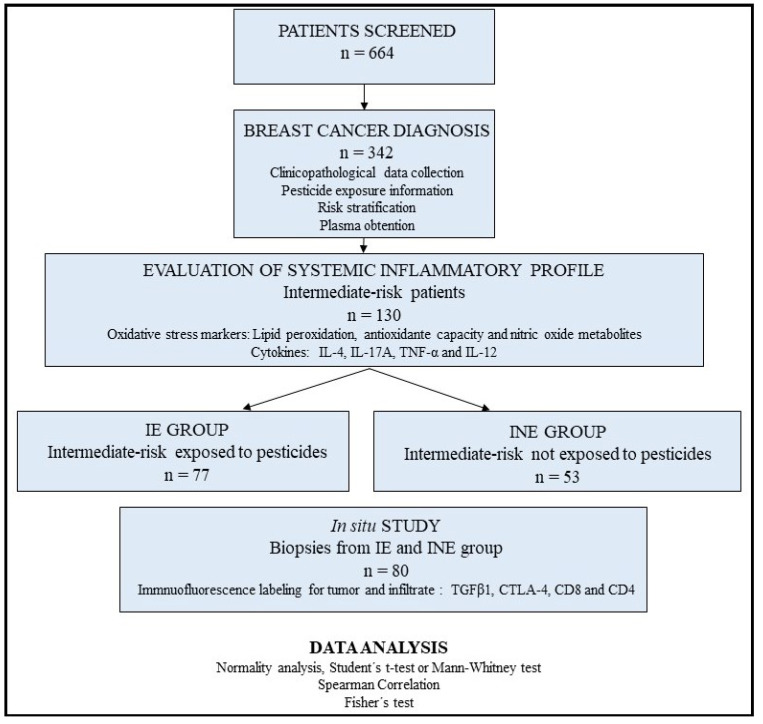
Study design. Six hundred sixty-four patients were screened. Three hundred forty-two were diagnosed with breast cancer. Based on risk stratification and pesticide exposure data, 130 patients were included and categorized into intermediate-risk exposed to pesticides (*n* = 77) and intermediate-risk unexposed to pesticides (*n* = 53). The in situ analysis was randomly selected by lot (*n* = 80).

**Figure 2 cancers-14-05199-f002:**
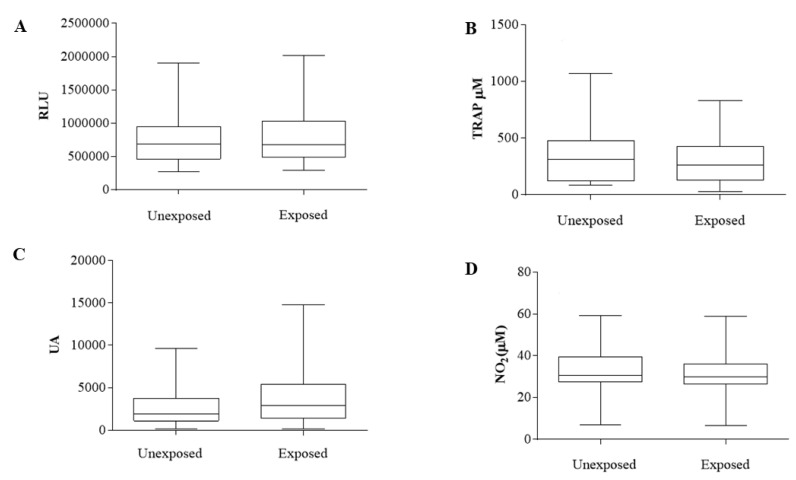
Oxidative stress analysis of plasma samples of exposed and unexposed breast cancer patients from intermediate-risk stratification. (**A**) Plasma hydroperoxide levels, measured in RLU = relative light units. (**B**) Total plasma antioxidant capacity (TRAP), measured in nM of Trolox. (**C**) Oxidation index in arbitrary units. (**D**) Levels of nitric oxide metabolites, measured in μM. Results are represented in box plots (min–max). Analyses were performed with GraphPad Prism 7.0 (GraphPad Software. San Diego, CA, USA).

**Figure 3 cancers-14-05199-f003:**
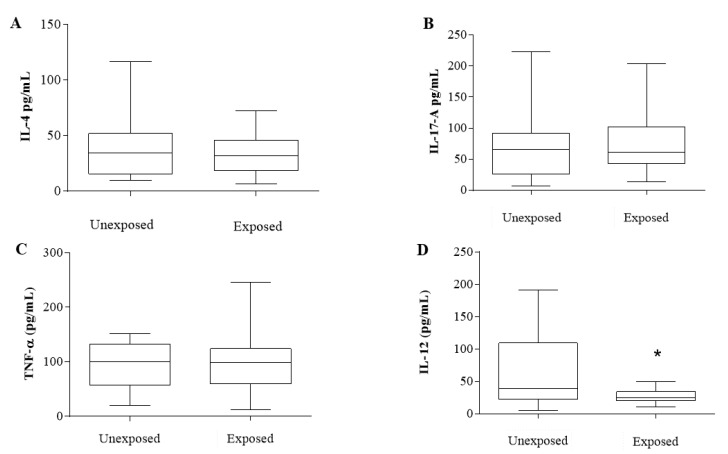
Cytokine profile of exposed and unexposed breast cancer patients from intermediate-risk stratification. (**A**) Interleukin 4. (**B**) Interleukin 17-A. (**C**) TNF-α. (**D**) Interleukin 12. All measured in pg/mL. Results are represented in box plots (min–max). * Indicates statistically significant difference (*p* < 0.05). Analyses were performed with GraphPad Prism 7.0 Software (San Diego, CA, USA).

**Figure 4 cancers-14-05199-f004:**
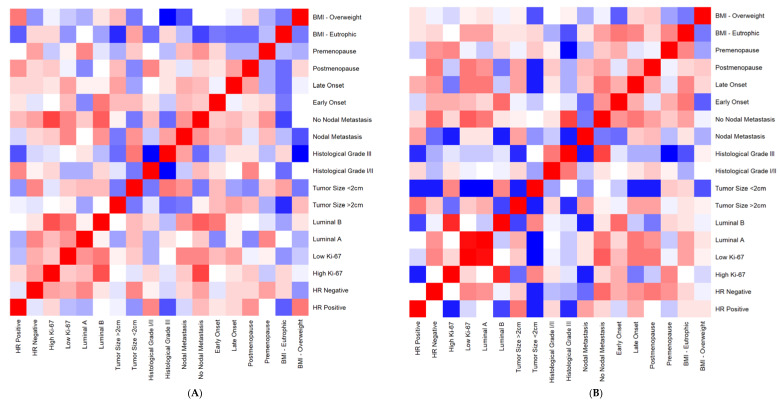
Heatmap: correlation analysis of clinicopathological data according to systemic levels of IL-12 of exposed and unexposed breast cancer patients from intermediate-risk stratification. (**A**) Exposed group. (**B**) Unexposed group. The analysis was performed with R programming language (R Development Core Team. 2021). Red squares indicate positive correlations. Blue squares indicate the negative ones. A more intense color indicates a stronger correlation.

**Figure 5 cancers-14-05199-f005:**
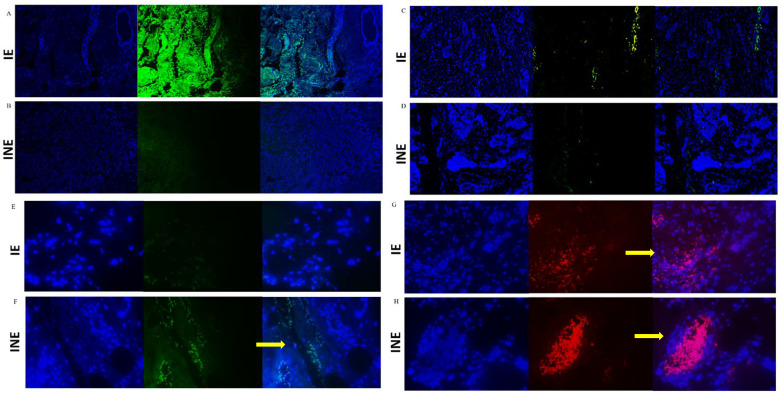
Immunostaining of TGF-β1 (**A**,**B**), CTLA-4 (**C**,**D**), CD8+ lymphocytes (**E**,**F**), and CD4+ lymphocytes (**G**,**H**) in breast tumor biopsies from intermediate-risk patients exposed (IE) or unexposed (INE) to pesticides. Labeling was evaluated in breast tumors and infiltrating leukocytes (400×). The images sequentially represent (horizontal view) DAPI labeling, marker labeling, and the merge of DAPI + marker. The resulting images were merged in ImageJ to generate the final images. For all images: immunostaining in green for Alexa Fluor (positive staining), red for Texas Red (positive staining), and blue for DAPI (negative counterstaining). The yellow arrows indicate the labeled areas.

**Figure 6 cancers-14-05199-f006:**
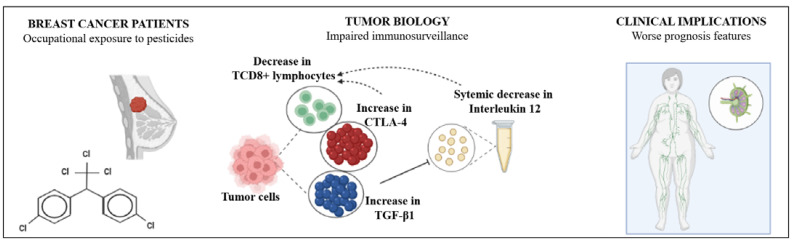
Pesticide-induced immune deregulation may worsen the prognosis in women with breast cancer from the intermediate risk of death and recurrence. Women diagnosed with breast cancer occupationally exposed to pesticides exhibit several significant alterations in the tumor microenvironment, affecting its immunosurveillance. Decreased TCD8+ lymphocyte infiltrate and increased expression of negative regulators of tumor-driven immune responses such as TGF-β and CTLA-4 are found in tumors from pesticide-exposed women compared to the unexposed ones. In addition, reduced levels of circulating interleukin 12 are reported, reducing the anti-tumor arsenal of pesticide-exposed women. This immunocompromised scenario may result in a worse clinical prognosis since women from the exposed group have more metastasis than the unexposed ones.

**Table 1 cancers-14-05199-t001:** Criteria for risk stratification of recurrence and death of patients diagnosed with breast cancer (based on Goldhirsch et al. [4]).

Low risk	Negative lymph nodes and all the following criteria:
	pT under 2 cm;
	Histological grade 1;
	ER or PR positive;
	HER-2 negative;
	Molecular subtype luminal A; and
	Age equal to or above 35 years old.
Intermediate risk	Negative lymph nodes and at least one of the following criteria:
	pT higher than 2 cm; or
	Histological grade 2–3; or
	ER or PR negative; or
	Molecular subtype luminal B (HER-2 negative); or
	Age under 35 years old; or yet
	1 to 3 affected lymph nodes if ER and PR positive.
High risk	4 or more positive lymph nodes; or
	Lymph nodes negative with ER. PR and HER-2 negative. pT higher than 2 cm; or
	Lymph node negative. pT higher than 1 cm and HER-2 positive.

pT: tumor size; ER: estrogen receptor; PR: progesterone receptor; HER-2: human epidermal growth factor receptor 2.

**Table 2 cancers-14-05199-t002:** Conditions of immunofluorescence reactions and antibody specifications.

Antibody	Clone	Reactivity	Titration	Incubation
Anti-TGF-β	Monoclonal antibody	Mouse anti-human	1:300	2 h in dark and humid chamber at room temperature
Alexa Fluor 488 ^a^	Superclonal recombinant secondary antibody	Goat anti-mouse IgG	1:500	1 h in dark and humid chamber at room temperature
Anti-CTLA-4	BNI3 monoclonal antibody	Mouse anti-human CD152	1:1000	overnight in dark and humid chamber at 4 °C
Alexa Fluor 488 ^b^	Superclonal recombinant secondary antibody	Goat anti-mouse IgG	1:1000	1 h in dark and humid chamber at room temperature
Anti-CD4	Monoclonal antibody	Mouse anti-human	1:1000	overnight in dark and humid chamber at 4 °C
Anti-CD8	Monoclonal antibody	Mouse anti-human	1:1000	overnight. in dark and humid chamber at 4 °C
Texas Red ^c^	Superclonal recombinant secondary antibody	Goat anti-mouse IgG	1:1000	2 h in dark and humid chamber at room temperature

^a^ For TGF-β1 labeling. ^b^ For CTLA-4 labeling. ^c^ For CD4 and CD8 labeling.

**Table 3 cancers-14-05199-t003:** Comparative analysis of clinicopathological data of exposed and unexposed breast cancer patients from intermediate-risk stratification.

Variable	Group	Category	%	*p* Value
Estrogen receptor	Exposed	Negative	7.79	>0.05
		Positive	92.21	
	Unexposed	Negative	5.88	
		Positive	94.11	
Progesterone receptor	Exposed	Negative	44.16	>0.05
		Positive	55.84	
	Unexposed	Negative	45.09	
		Positive	54.9	
KI67 (%)	Exposed	<14	41.09	>0.05
		>14	58.9	
	Unexposed	<14	45.83	
		>14	54.16	
Molecular subtype	Exposed	Luminal A	38.96	>0.05
		Luminal B	57.14	
		Triple negative	3.9	
	Unexposed	Luminal A	43.13	
		Luminal B	47.05	
		Triple negative	9.8	
Tumor size (cm)	Exposed	<2	35.61	>0.05
		>2	64.38	
	Unexposed	<2	40	
		>2	60	
Histological grade	Exposed	I and II	81.33	>0.05
		III	18.66	
	Unexposed	I and II	80.39	
		III	19.6	
Lymphnodal metastases	Exposed	None affected	60	0.0353 *
		At least one affected	40	
	Unexposed	None affected	74	
		At least one affected	26	
Age at diagnosis	Exposed	<50	32.47	>0.05
		>50	67.53	
	Unexposed	<50	39.21	
		>50	60.78	
Menopause status at diagnosis	Exposed	No	27.27	>0.05
		Yes	72.73	
	Unexposed	No	35.29	
		Yes	64.7	
BMI	Exposed	Eutrophic	25	0.0478 *
		Overweight/obese	75	
	Unexposed	Eutrophic	38	
		Overweight/obese	62	

* *p* < 0.05. Chi-square analysis.

**Table 4 cancers-14-05199-t004:** TGF-β1 and CTLA-4 expression profile in tumor tissue and immunological infiltrate from tumor biopsies.

Marker	INE	IE	*p* Value ^Δ^
TGF-β1 tumor ^a^	70%	80%	NS
TGF-β1 infiltrate ^a^	40%	90%	<0.001 *
CTLA-4 tumor ^a^	40%	10%	<0.001 *
CTLA-4 infiltrate ^a^	70%	100%	<0.002 *
CD4 infiltrate ^b^	50%	60%	NS
CD8 infiltrate ^b^	50%	30%	0.0059 *

* Indicates *p* < 0.05. NS: not significant (>0.05). ^a^ Any positive labeling. ^b^ Positive labeling in at least 25% of observed lymphocytes. **^Δ^** Fisher’s exact test.

## Data Availability

Data will be available under request.

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
