# Peer review of "Occupational Exposure to Pesticides Affects Pivotal Immunologic Anti-Tumor Responses in Breast Cancer Women from the Intermediate Risk of Recurrence and Death"

_cancers, 2022, doi:10.3390/cancers14215199_

Round 1

Reviewer 1 Report

The manuscript entitled “Occupational Exposure to Pesticides Affects Pivotal immunologic Anti-Tumor Responses in Breast Cancer Women From the Intermediate Risk of Recurrence and Death” by Janaína Carla da Silva is a novel study having crucial data compilation from human breast cancer patients. However, there are a few queries that need to be addressed before making any final decision.

1.       How did the author isolate intermediate-risk patients from the total patients? Do these patients give pesticides before or after the cancer occurrence?

2.       Out of 130 patients used for the study, how many of them had ER+, PR+, and triple-negative type breast cancer? The final data which you represented is very crucial and dependent on the type of breast cancer patients you choose for the study.

3.       Oxidative analysis showed no changes in the unexposed and exposed patients. How would you explain this analysis? It is well known that cancer increases oxidative stress levels; however, your data showed the opposite results. Please give a reason.

4.       What is the black box represent in the heatmap (figure 4)?

5.       Please club all four- Figure 5 altogether and use the yellow highlighted arrow to represent what you show in a particular slide.

Author Response

REVIEWER COMMENTS

Reviewer #1

The manuscript entitled “Occupational Exposure to Pesticides Affects Pivotal immunologic Anti-Tumor Responses in Breast Cancer Women From the Intermediate Risk of Recurrence and Death” by Janaína Carla da Silva is a novel study having crucial data compilation from human breast cancer patients. However, there are a few queries that need to be addressed before making any final decision.

  1. How did the author isolate intermediate-risk patients from the total patients? Do these patients give pesticides before or after the cancer occurrence?

Thank you for your comment. The intermediate-risk patients were isolated from the total patients by using the following criteria reported in Table 1.

Negative lymph nodes and at least one of the following criteria:

pT higher than 2 cm; or

Histological grade 2 - 3; or

ER or PR negative; or

Molecular subtype luminal B (HER-2 negative); or

Age under 35 years old; or yet

1 to 3 affected lymph nodes. if ER and PR positive.

These patients were continuously exposed to pesticides, before cancer diagnosis and after, since they are rural workers and most of them did not stop their activities after diagnosis. We added this information in the Methods section. Thank you.

  1. Out of 130 patients used for the study, how many of them had ER+, PR+, and triple-negative type breast cancer? The final data which you represented is very crucial and dependent on the type of breast cancer patients you choose for the study.

Thank you for your comment. As shown in Table 2, the compared groups had similar profiles concerning this issue (p>0.05). About 92% of exposed and 94% of unexposed had ER positive tumors, and about 55% of patients had PR positive tumors in both groups. Also, most of patients from both groups had Luminal tumors.

  1. Oxidative analysis showed no changes in the unexposed and exposed patients. How would you explain this analysis? It is well known that cancer increases oxidative stress levels; however, your data showed the opposite results. Please give a reason.

Thank you for your comment.

We agree that oxidative stress is a condition in breast cancer. We have previously characterized oxidative stress levels in breast cancer patients (doi: 10.1007/s00262-012-1283-8) and found similar levels as those reported in the present study. Also, this previous study demonstrated that such levels are higher than oxidative stress levels observed in healthy women without breast cancer (controls).

It is important to observe that our data compared two groups that already had a breast cancer diagnosis and showed no difference in oxidative stress levels when comparing both exposed vs. unexposed.

  1. What is the black box represent in the heatmap (figure 4)?

 Thank you for your comment. This box was removed.

  1. Please club all four- Figure 5 altogether and use the yellow highlighted arrow to represent what you show in a particular slide.

Thank you for your suggestion, we changed the figure as requested.

Reviewer 2 Report

Dear Authors:

The manuscript "Occupational Exposure to Pesticides Affects Pivotal Immunologic Anti-Tumor Responses in Breast Cancer Women From the Intermediate Risk of Recurrence and Death." by Da Silva et al has demonstrated that occupational exposure to pesticides changes the molecular behavior of disease and should be considered for intermediate risk stratification assessment in breast cancer patients. I have just a few suggestions.

1. Some background information or reference is missing: Please add more background information about roles of inflammation, obesity and ROS in breast cancer. (Please cite: 1. An Epigenetic Role of Mitochondria in Cancer. Cells 202211, 2518. https://doi.org/10.3390/cells11162518 

2. Advances in the Prevention and Treatment of Obesity-Driven Effects in Breast Cancers. Front Oncol. 2022 doi: 10.3389/fonc.2022.820968.

3. Mitochondrial mutations and mitoepigenetics: Focus on regulation of oxidative stress-induced responses in breast cancers. Semin Cancer Biol. 2022 Aug;83:556-569. doi: 10.1016/j.semcancer.2020.09.012.)

2. The manuscript needs linguistic improvement

3. If it it possible, please add more figures about potential mechanisms 

Best,

Author Response

Reviewer #2

The manuscript "Occupational Exposure to Pesticides Affects Pivotal Immunologic Anti-Tumor Responses in Breast Cancer Women From the Intermediate Risk of Recurrence and Death." by Da Silva et al has demonstrated that occupational exposure to pesticides changes the molecular behavior of disease and should be considered for intermediate risk stratification assessment in breast cancer patients. I have just a few suggestions.

  1. Some background information or reference is missing: Please add more background information about roles of inflammation, obesity and ROS in breast cancer. (Please cite: 1. An Epigenetic Role of Mitochondria in Cancer. Cells202211, 2518. https://doi.org/10.3390/cells11162518 

Thank you for your suggestions. We added all mentioned references in the discussion section.

  1. Advances in the Prevention and Treatment of Obesity-Driven Effects in Breast Cancers. Front Oncol. 2022 doi: 10.3389/fonc.2022.820968.

Thank you for your suggestions. We added all mentioned references in the discussion section.

  1. Mitochondrial mutations and mitoepigenetics: Focus on regulation of oxidative stress-induced responses in breast cancers. Semin Cancer Biol. 2022 Aug;83:556-569. doi: 10.1016/j.semcancer.2020.09.012.)

Thank you for your suggestions. We added all mentioned references in the discussion section.

  1. The manuscript needs linguistic improvement

Thank you for your suggestions. The manuscript was revised by a native English speaker. 

  1. If it it possible, please add more figures about potential mechanisms 

We added a mechanistic Figure in the manuscript (Figure 6). Thank you.

Round 2

Reviewer 2 Report

strongly suggestion for publication